# Exploring Biases in Facial Expression Analysis using Synthetic Faces

**Ritik Raina**[1]   **Miguel Monares**[1]   **Mingze Xu**[1]   **Sarah Fabi**[2]   **Xiaojing Xu**[1]
**Lehan Li**[1]   **William Sumerfield**[1]   **Jin Gan**[1]   **Virginia R. de Sa**[1]
[1]University of California, San Diego   [2]University of Tübingen
{rraina,mmonares,m6xu,xix068,l8li,wsumerfi,j6gan,desa}@ucsd.edu
sarah.fabi@uni-tuebingen.de

## Abstract

Automated facial expression recognition is useful for many applications, but models are often subject to racial biases. These racial biases may be hard to reveal due to the complexity and opacity of the complex networks needed for state-of-the-art performance. Racial biases are also hard to demonstrate due to the inability to fully match facial expressions across real people. In this paper we use artificially created faces where facial expression can be carefully manipulated and matched across artificial faces with different skin colors and different facial shapes. We show that several public facial expression models appear to have racial biases. This work is an important step towards the eventual goal of understanding the basis of these biases and removing them from facial expression models.

## 1   Introduction

Automated facial expression recognition (FER) is useful for many applications including monitoring: student engagement and boredom, consumer interest and happiness, customer distress, and pain levels in patients who are unable to communicate either due to age, motor disabilities, or cognitive state.

It is important that recognition of facial expressions be fair and consistent in recognizing expressions in people of different races and skin colors. One challenge in achieving model fairness in deep learning FER models is the complexity of deep learning processes. Due to the black-box nature of these models, it is difficult to understand how the FER models digest and interpret facial features to score emotion. Therefore, researchers often struggle to identify the weaknesses of FER models that prompt disparate performance due to racial biases. Such biases may arise due to the under-representation of specific populations or due to difficulties in detecting relevant features that vary with different skin color or facial features.

Another challenge in achieving model fairness is the difficulty of obtaining facial expression datasets with sufficient quality, quantity, and diversity. The traditional means of obtaining data for facial expression datasets is expensive and cumbersome. Also, when using pictures or videos of real people making either natural or posed facial expressions, it is not possible to exactly control all aspects to ensure, for example, that a White face is making exactly the same expression (and thus should be scored in the same way) as a Black face.

Recently, Fabi et al. [6] used artificially generated faces to explore racial bias in pain-related facial expressions in a specific computer vision pain-estimation model [16]. They revealed that, while the network was able to respond largely monotonically and fairly linearly to increases in activation of facial AUs, it was subject to different biases, and gains for different skin colors and races. Importantly, the biases and gains were not solely better for the faces of the majority race and skin color.

In this work, we generate and use new artificially generated faces made by the FaceGen Modeller as a means to explore racial biases in several publicly available computer vision facial expression

models. The advantages of synthetic face images are that we can control facial parameters (such as race and skin color), while holding modeled facial muscle activation constant, to isolate how different manipulations affect the different models. This work serves as an elucidation of the problem and some of the dependencies which will enable future work towards removing these biases from facial expression models. This work may also help inform any sensory components to human bias in FER when observing people of color.

## 2 Methods

In this section, we describe our synthetic face dataset and explore what results facial expression models produce with controlled manipulations to skin color, facial morphology, and facial muscle activation.

### 2.1 FaceGen Synthetic Image Dataset

Our work uses artificially generated facial images made through the FaceGen Modeller software. FaceGen Modeller is a programming tool that can generate synthetic faces based on different manipulations you provide such as race, facial expression, coloring, gender, etc. Our dataset consists of four sets of "races": "European", "European features with dark skin color" or "European Black", "African", and "African features with light skin color" or "African White". We used the FaceGen Modeller to generate random male faces using the FaceGen "African" and "European" racial settings. (Male faces were used as they appeared to be more realistic than the female FaceGen faces). For the original "African" and "European" faces, we did not change any parameters under the category "modify". For the other two sets, the skin shade of the faces were set between 2.5 (lighten) to -3.0 (darken). For the set of "European Black", we took each European face and set the skin shade parameter towards -3 adjusting the brightness until the face looked most natural. Similarly, for the set of "African White", we took each African face and set the skin shade parameter toward 2.5 until it looked as natural as possible. "African White" and "European Black" are thus two artificial categories we created to observe how coloring alone (with no change in face shape) affected the models. For the remainder of the paper, we refer to the four categories as "races".

The expressions of the synthetic faces were constructed via manipulating the facial action unit (AU) activation levels. The Facial Action Coding System (FACS) [5] codified the AUs as anatomically-based components of facial muscle movements underlying perceivable facial movements. After modifying the races and colors, we manipulated the facial expression with ten AUs (4: Brow Lowerer, 6: Cheek Raiser, 7: Lid Tightener, 9: Nose Wrinkler, 10: Upper Lip Raiser, 12: Lip Corner Puller, 20: Lip stretcher, 25: Lips part, 26: Jaw Drop, 43: Eyes Closed), each of which was activated to six levels (0%, 20%, 40%, 60%, 80%, 100%) separately (except for AU43: Eyes Closed which was only set to 0% and 100%). We chose to manipulate these AUs individually in order to isolate the effects that would be made on each of our models. This step was applied to the four racial sets in the exact same way, leading to 224 images in total. Fig.1 shows rows of synthetic faces set with the same expressions, except having differing facial morphologies and skin color.



Figure 1: Synthetic face images used for experimental evaluations. Each row shows the same expression for each of African, African White, European, and European Black settings respectively. (First row) This row involves the activation of AU7-Lid Tightener (100% activation) with rest of the AU activations to be set at 0%. (Second row) In this row AU10-Upper Lip Raiser is activated to 100% for all face settings. (Third row) In this row AU25-Lips part is activated to 100% for all face settings. (Fourth row) In this row AU26-Jaw Drop is activated to 100% for all face settings.

## 2.2 Extended MTL Model for Pain-Estimation

The first model we investigate, follows on the work of [6] to investigate the first-stage of the extended multi-task learning (MTL) pain estimation neural network of Xu et al. [16], referred as the Pain-Estimation model from here onward. This model was trained and achieved state-of-the-art accuracy on the UNBC-McMaster Shoulder Pain dataset [9]. The UNBC-McMaster dataset is publicly available and comprises of 200 face videos from 25 patients with shoulder pain as they performed movements of their shoulder resulting in differing levels of pain. From these videos, each frame was labelled with 11 facial action unit (AU) intensities, and a corresponding PSPI value, and 66 AAM landmarks. The PSPI score [12] is a pain metric computed from a distinct set of AU intensities that are pain-related. The first stage of this three-staged model is a VGGFace network [11] pre-trained on classifying 2622 faces of celebrity icons, most of which are racially white. This network was then additionally trained on the UNBC-McMaster Shoulder Pain dataset to detect pain-related AUs (4, 6, 7, 10, 12, 20, 25, 26, 43) and the PSPI intensity of a facial image. The extended MTL model had two other stages that took the output of stage one and predicted whole-video segment pain scores, but our experiments only utilize the first stage of this model. This model is publicly available.

## 2.3 Multi-task EfficientNet-B2 Emotion Classification

The second model that we evaluate, Multi-task EfficientNet-B2, follows the work of [13] on the emotion and engagement classification in an online learning setting. This state-of-the-art model follows a two-stage training structure of pre-training on face detection and fine-tuning on emotion classification. The face detection stage of this model is a pre-trained VGGFace network and uses tracking & clustering techniques to extract facial sequences. Then, the model is fine-tuned on the largest publicly available facial emotion dataset, AffectNet [10] - a collection of 287,651 images annotated with eight basic emotions (anger, contempt, disgust, fear, happiness, neutral, sadness, and surprise). The training set of AffectNet is known to be biased, as 64.4% of the faces are White (of 7 total races) [3]. We included this model in our evaluation because it achieved the greatest performance for facial emotion recognition on AffectNet, obtaining the highest known accuracy on both 8-emotion and 7-emotion classification. This model is publicly available.

## 2.4 Intel OpenVINO Emotion Recognition

Intel's Open Visual Inference and Neural Network Optimization (OpenVINO) library is a series of toolkits that provides effective deep learning neural networks on various tasks such as object, detection, segmentation, and estimation [1]. Their emotion recognition model performs on top of their underlying face detection model which produces coordinates over the bounding box surrounding the detected face. The underlying network features a MobileNet backbone [7] that utilizes depthwise separable convolutions as a measure for running neural networks efficiently on mobile devices. The emotion recognition has been trained and evaluated with only five emotions ('Neutral', 'Happy', 'Sad', 'Surprise', 'Anger') from the AffectNet dataset. The architecture of the emotion recognition model is that of a fully connected neural network that outputs a softmax probability across the five emotions detected. This model is publicly available.

# 3 Experiments

## 3.1 Experiment 1: Racial biases in facial action unit estimation.

**Synthetic image transformations for AU estimation evaluation.** In this experiment, we explored the task of facial action unit estimation on the synthetic face dataset we generated. For evaluating on the Pain-Estimation model [16], we utilize the MTCNN (multitask CNN) [18] face detector to detect the faces off the synthetic images. This aspect was troublesome for the dark-skinned faces as it would sometimes not detect them at all, or mistake the facial bounding box coordinates; unlike the correct detections made on the light-skinned faces. In order to mitigate this issue, we cropped the dark-skinned faces with the complementing light-skinned face bounding box coordinates. The bounding box surrounding the detected face is extended by a factor of 0.3 before cropping. The input images are reshaped to 256×256, center-cropped to 224×224 and their color channels were adjusted to that of the VGGFace network.

**A bias in dark-skinned vs light-skinned facial AU activation estimation is prevalent.** For the nine AU estimations that are considered in the Pain-Estimation model, we sampled facial images with those specific AUs being manipulated, across the four sets of races mentioned earlier. We ran model inference on those images and explored performance disparities between these racially diverse faces on AU estimation. We then analyzed the results by running paired t-tests based on skin color. Table 1 shows results on the test-statistics and p-values returned for faces with the same facial expressions and facial shapes, but differing skin colors. For these experiments, the Bonferroni corrected alpha is 0.05/18 or 0.0027. These statistics are used to test for AU estimation inequalities in faces that only differ by color. For AU20, AU25, and AU43, we observe p-values which are not above our threshold for significance when comparing estimations in African vs. African-White faces (AU20: ($t$=1.59, $p$=0.1728), AU25: ($t$=-0.71, $p$=0.5075), AU43: ($t$=1.853e-3, $p$=0.9988)). Similarly, AU4, AU6, AU12, AU26, and AU43 revealed no significant differences in AU activations for European vs European-Black faces (AU4: ($t$=-0.71, $p$=0.5117), AU6: ($t$=-4.52, $p$=0.0062), AU12: ($t$=2.94, $p$=0.0322), AU26: ($t$=-2.73, $p$=0.0413), AU43: ($t$=-0.71, $p$=0.6062)). However, the other AUs demonstrated disparate predictions between dark-skinned and light-skinned versions of the same faces. For example, in Fig.2, the model produced higher AU6 and AU10 activations for African faces than the African-White faces. Despite the fact that the facial morphologies in the African/African-White faces were the same, the sole manipulation of skin color resulted in disparate performance, revealing a color bias during AU activation estimation within the model.

| Paired t-test on AU activation estimations | | | | |
|---|---|---|---|---|
| | African vs. African-White | | European vs. European-Black | |
| Action Unit | test stat. | uncorrected p-value | test stat. | uncorrected p-value |
| 4 | -6.94 | 9.5077e-4 | -0.71 | 0.511713 |
| 6 | 99.48 | 1.9460e-9 | -4.52 | 0.0062 |
| 7 | -9.73 | 1.9441e-4 | 12.13 | 0.000067 |
| 10 | 7.21 | 7.9851e-4 | -5.62 | 0.0024 |
| 12 | 13.83 | 3.5373e-5 | 2.94 | 0.0322 |
| 20 | 1.59 | 0.1728 | 17.86 | 0.000010 |
| 25 | -0.71 | 0.5075 | 18.15 | 0.000009 |
| 26 | -5.18 | 3.5034e-3 | -2.73 | 0.0413 |
| 43 | 1.853e-3 | 0.9988 | -0.71 | 0.6062 |

Table 1: Paired t-tests on AU activation estimations made by the Pain-Estimation model. Each test was based on the model's evaluations on the individual AU manipulations (on the left column) with respect to the four racial sets.

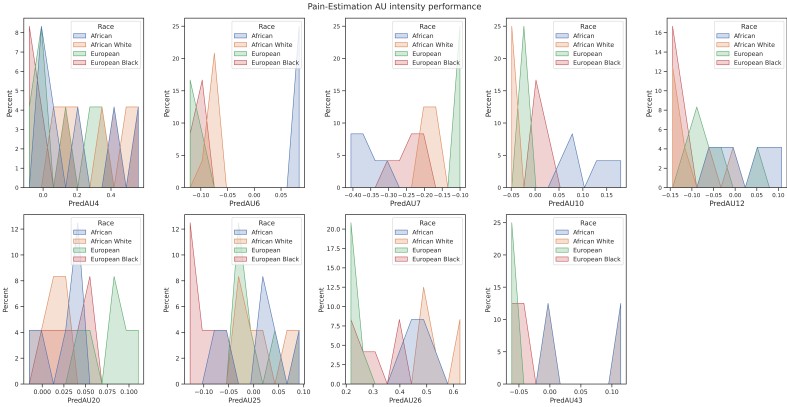

Figure 2: Histograms comparing the percentages of predicted AU estimation distribution in the Pain-Estimation model. Each predicted AU plot distribution pertains to the model's AU activation predictions on those synthetic images that had that AU manipulated. Each plot shows the AU estimation predictions being made to the four sets of racially diverse images - African (Blue)/African-White (Red)/European (Orange)/European-Black (Green).

## 3.2 Experiment 2: Racial biases in facial emotion classification.

**Synthetic image transformations for emotion classification evaluation.**

In this experiment, we explored the bias of several facial emotion recognition models using the same synthetic dataset as Experiment 1. During model inference on the Multi-task EfficientNet-B2 model, the input images were represented in RGB format, cropped to 224×224, and normalized with ImageNet [2] mean and standard deviation values. For the evaluation on the Intel OpenVINO emotion recognition model, the synthetic images were resized to 64×64 and their color channels were ordered in BGR format.

**Classification intensities showed polarizing biases between negative and positive emotions.** The Multi-task EfficientNet-B2 emotion classification model was evaluated on the entire synthetic face dataset and showed a greater bias surrounding negative emotion classification (possibly due to our synthetic dataset having more AUs related to negative emotions), than positive. Fig.3 (Top) shows that the predicted emotion intensities vary with the skin colors. We also see that the face morphologies of the synthetic faces impact the emotion intensities. In addition to Fig.3, we performed paired t-tests between African vs. African-White, African-White vs. European, African vs. European-Black, and European vs European-Black faces. The Bonferroni corrected alpha for these tests is 0.05/40, or 0.00125. The "disgust" emotions rated higher intensities for African-White faces than European faces ($p$=2.84e-8). Although both subsets had similar skin colors, their differing facial morphologies prompted differences in emotion intensity predictions. On the other hand, European and African-White faces exhibited higher "fear" intensities than the European-black and African counterparts respectively ($p$=7.03e-7 on European vs. European-Black and $p$=6.78e-11 on African vs. African-White).

To delve deeper into the bias with negative emotion classification, we evaluated the model on faces that had 0% activations across all AUs for "disgust", "contempt", "fear", "sadness", and "anger" emotions. As shown in Fig.3 (Bottom), the African-White faces conveyed higher "disgust" intensities as compared to the African faces showing a clear color bias; however, the European and European-Black faces were seen to be having similar intensities on "disgust" and "sadness". Another case of color bias is also seen in "fear" where European and African-White faces yield higher intensity.

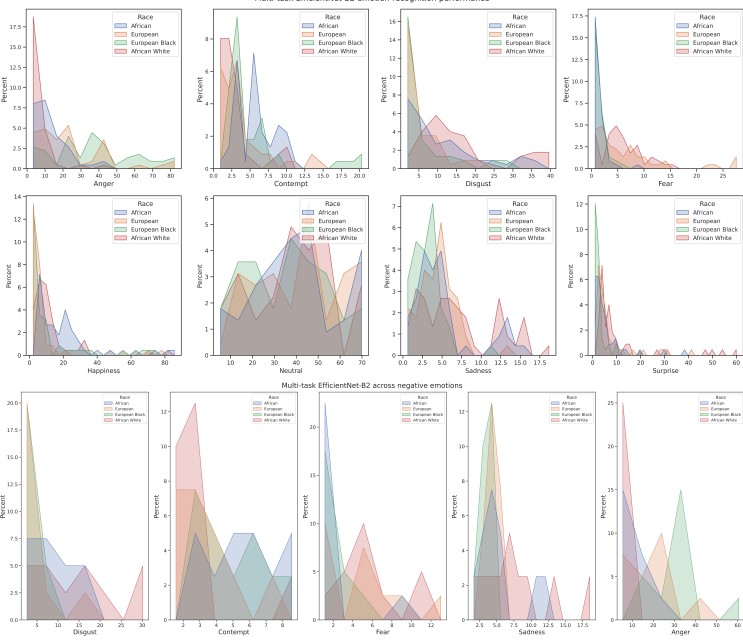

Figure 3: (Top two rows) State-of-the-art AffectNet model (Multi-task EfficientNet-B2) performance on emotion intensity classification on the synthetic face dataset across all racial sets and AU activation manipulations. The emotions received are "anger", "contempt", "disgust", "fear", "happiness", "sadness", "neutral", and "surprise". The intensities range from (0%-100%). (Bottom row) Histograms reflecting the Multi-task EfficientNet-B2's predictions on synthetic faces that had 0% activations on all AUs. These histograms are to show the effect over the four sets of racial backgrounds, across the negative emotions of "disgust", "contempt", "fear", "sadness", and "anger".

Similar color biases can be seen in the OpenVINO emotion recognition model - in which disparities amongst negative and positive emotion intensities are exhibited. Fig.5 shows the probability distribution across the "happy" (positive) and "sad" (negative) emotions being represented from the synthetic faces which had 0% activations over all its AUs. We ran paired t-tests on the model's emotion clasification predictions over African vs. African-White and European vs. European-Black faces as shown on Table 2. These tests had a Bonferroni corrected alpha of 0.05/10 or 0.005. The table shows that there was a difference in the model's prediction of the "happy" emotion for the faces with "African" morphology where the African-White faces had higher "happy" intensity than African ($t$=-5.90, $p$=0.0002). Additionally, African-White faces had higher intensity values in comparison to the African faces on the "sad" emotion. These differences were not found to be significant between European and European-Black faces.

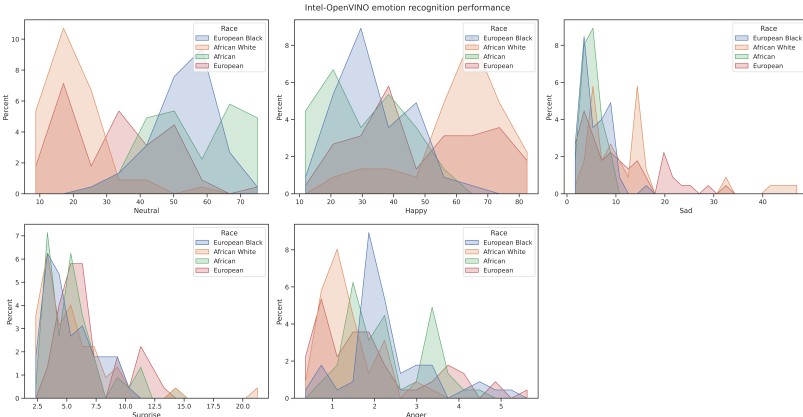

Figure 4: Intel OpenVINO performance over the synthetic face dataset and their emotion probability evaluations.

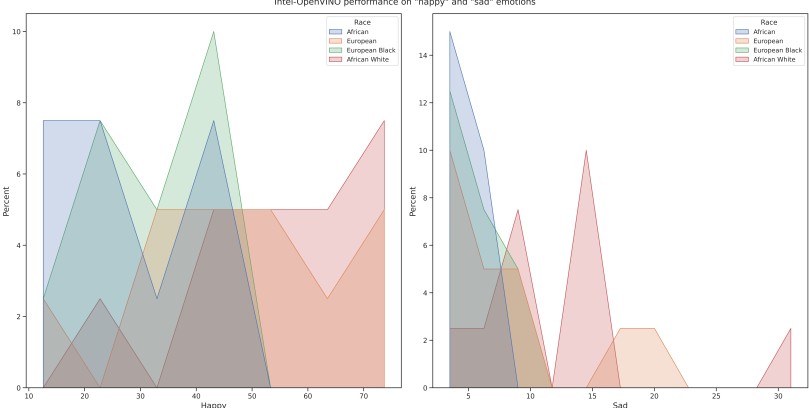

Figure 5: Histogram plots showing OpenVINO's performance on emotion intensity classification on the synthetic faces that had 0% activations over all AUs.

| | Paired t-test on emotion intensity prediction | | | |
|---|---|---|---|---|
| | African vs. African-White | | European vs. European-Black | |
| Emotion | test stat. | uncorrected p-value | test stat. | uncorrected p-value |
| Neutral | 8.74 | 0.000011 | -2.92 | 0.0169 |
| Happy | -5.90 | 0.0002 | -2.06 | 0.0696 |
| Sad | -3.27 | 0.0096 | 1.28 | 0.2299 |
| Surprise | 0.76 | 0.4625 | 1.03 | 0.3305 |
| Anger | 4.94 | 0.0008 | -0.42 | 0.6851 |

Table 2: Paired t-tests on emotion classification made by OpenVINO. The tests are based on the synthetic faces with 0% activation across all AUs.

# 4 Related Work

Inherent biases in facial expression recognition models have been studied previously by model fairness and computer vision researchers. Further studies on biased models have been observed with changes to gender, age, and facial characteristics (e.g. facial hair, clothing) as well [4], [3], [15]. Xu et al. [15] discusses how current facial datasets do not portray the same diverse distribution of facial attributions as seen across the human population. Catena et al. [3] attributed their findings of model gender bias to discrepancies found in training datasets. Furthermore, Domnich et al. [4] observed that annotation biases in expression datasets were a result of human errors and resulted in reductions in accuracy and fairness in the resulting trained model.

Our work is closely related to Shadmi et al. [14] and Kortylewski et al. [8] where synthetic facial images are used to identify and reduce negative effects of biased datasets on landmark detection and facial expression recognition. The work of Kortylewski et al. [8] showed differences in model accuracies from retraining on synthetic faces. They portray how synthetic face generation allows for models to extrapolate to more real-life facial expressions across diverse facial population data. Although our current work has only started to explore the biases that exist in open-source models, we intend to keep using synthetic faces as a means of further understanding and, in the future, debiasing facial expression analysis models.

# 5 Discussion

We developed a synthetic face image dataset rich with diversity using manipulations of skin-color, action unit activation, and facial morphology by race. In our investigation of racial biases in public facial expression analysis models, synthetic images were a useful tool for exposing bias in FER models due to color and facial morpohology. Using the FaceGen Modeller, we manipulated facial identity and race to create faces that differed across skin colors and facial morphologies. Next, we controlled facial expressions by manipulating activation intensities across ten AUs. We performed two experiments on analyzing the predictions being made by (1) an AU estimation model and (2) two emotion recognition models. We observed color biases being instantiated by the Pain-Estimation model, where certain AU estimations had higher predictions for certain skin colors. In synthetic faces where all facial parameters were equal except color, there is an apparent gap in detected intensity between several AUs. However, we did not observe that one skin color necessarily performed better or worse than the other in terms of performance or intensity estimation. This suggests that, despite the racial imbalances of the training datasets, model bias is not solely representative of dataset representation biases. Additionally, we observed biases due to differing facial morphologies in the public FER models. Specifically, we discovered gaps in intensity level during emotion classification between faces that were similar in skin color, but differed in facial features. The investigations presented in this paper expose the racial bias present in several public facial emotion analysis models. Exploring such biases using artificially generated faces demonstrates the value of using synthetic data to carefully manipulate facial features in order to deliberately examine underlying biases that exist in FER models. In future work, we will leverage synthetic images to help identify and mitigate the root of racial biases in FER models possibly by incorporating estimations of uncertainty [17].

# 6 Acknowledgements

This work was supported by UC San Diego Social Sciences (Advancing Racial Justice Award) and the Sanford Institute for Empathy and Compassion (Center for Empathy and Technology Award). We thank the FaceGen Modeller team for creating the software used to create the synthetic faces.

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
