# OpenReview forum: "Exploring Biases in Facial Expression Analysis using Synthetic Faces"
_NeurIPS.cc/2022/Workshop/SyntheticData4ML — Neurips 2022 SyntheticData4ML_

### Official Review · Reviewer_vmGp · 2022-10-17
**In this paper, authors created synthetic face image dataset and investigated the racial biases by considering expressions of different races.**

**Rating:** 8
**Confidence:** 3

**Review:**

 In this paper authors synthesized faces to remove racial bias and show that existing dataset consist of racial biases.

Proposed work uses FaceGen Modeller software to generate facial images artificially. Dataset considers four sets of "races": “European”, “European features with dark skin color” or "European Black", “African”, and “African features with light skin color” or "African White".

The expressions of the synthetic faces were manipulated. Then existing models are used for Pain-Estimation, Emotion Classification, and Emotion Recognition.

Experiments demonstrate those biases; For example, there is a gap between action unit intensities where all parameters were equal except color in public datasets. In another example the author demonstrates the gaps in intensity level during emotion classification between faces that were similar in skin color but differed in facial features.

Overall approach is very interesting, experiments are comprehensive, and conclusive.

Few Suggestions to add at the time of camera ready (depending on the space):
- Related work is not clear, current approaches and background need to be separated from Section 2.
- How others evaluated their dataset, are they using the same races, maybe include a few lines around it.
- If we remove the biases what will be its effect on overall evaluation metrics of datasets, any thoughts, examples of metrics, expected behavior?
- any known disadvantages of using the proposed approach?

---

### Official Review · Reviewer_37Jj · 2022-10-18
**This paper explores the biases of several public facial expression models using synthesized images**

**Rating:** 6
**Confidence:** 5

**Review:**

In this paper, the authors developed a synthetic face image dataset with intentional diversity of skin-color,
 and facial morphology.  The authors then investigate the biases of several public facial expression models.
They performed two experiments on analyzing predictions being made by (1) an AU estimation model and (2) two emotion recognition models. They observed color biases being instantiated by the Pain-Estimation model, where certain AU estimations had higher predictions for certain skin colors.  However, they did not observe that one skin color necessarily performed better or worse than the others
They discovered gaps in intensity level during emotion classification between faces that were similar in skin color, but differed in facial features.
These observations are very important for academia and that is why I believe this paper should be presented.
One key aspect of bias in face datasets is age. Majority of the face datasets are focused towards young people and it would be interesting to investigate the effect of age as well.

---

### Official Review · Reviewer_RaSy · 2022-10-18
**Nice analysis and dataset contribution which could be improved with additional comparisons and inclusion of related work**

**Rating:** 5
**Confidence:** 4

**Review:**

* The authors highlight the prevalence of racial biases in facial understanding tasks and attempt to quantify variance through synthesized datasets with a fair and balanced representation. The main contribution of the paper seems to be a framework to understand these biases and importantly, artificial datasets which can be used to evaluate performance.
* While the paper has clear structure and is easy to follow, there is less discussion of existing work on identifying biases and distribution pitfalls in models. Adding more relevant discussions of such datasets and approaches and comparing the proposed dataset to them would make the paper a stronger contribution.
* While an interesting investigation, the novelty of the work is limited and authors rely on off-the-shelf tools to generate the dataset with limited novelty in the generation itself. However, the analysis and applications of the dataset are appreciated.
* The overall writing and presentation of results could also be improved.
* Overall, it is an interesting investigation and nice contribution to propose an unbiased dataset to evaluate facial understanding tasks. However additional discussions and comparisons to other approaches and datasets in the area would make the insights much stronger.

---

### Meta-Review · Area_Chair_R7D2 · 2022-10-19

**Recommendation:** Accept